# Research on Spatial and Temporal Pattern Evolution and Driving Factors of County Carbon Emissions in Underdeveloped Regions: Gansu Province of Western China as an Example

Weiping Zhang [1], Peiji Shi [1,2,3,*] and Wanzhuang Huang [1]

[1] College of Geography and Environmental Science, Northwest Normal University, Lanzhou 730070, China
[2] Gansu Engineering Research Center of Land Utilization and Comprehension Consolidation, Northwest Normal University, Lanzhou 730070, China
[3] Key Laboratory of Resource Environment and Sustainable Development of Oasis, Lanzhou 730070, China
* Correspondence: shipj@nwnu.edu.cn; Tel.: +86-138-9366-5158

**Abstract:** This paper used the Gini coefficient, standard deviation ellipse, and spatial autocorrelation model to analyze the overall changes, regional differences, spatio-temporal evolution pattern, and clustering characteristics of carbon emissions in 87 counties in Gansu Province from 1997 to 2017, based on which driving factors of carbon emissions were detected using the geographic detector model, so as to provide a reference for promoting low-carbon green development and ecological civilization construction in Gansu Province. The empirical research results found that county carbon emissions in Gansu Province showed a "first urgent and then slow" upward trend, and the difference in carbon emissions level has a slightly decreasing trend, and there are significant regional differences. Compared with other regions, the difference in county carbon emissions level in the Longzhong region has a smaller decline. Meanwhile, the county carbon emissions show spatial differentiation characteristics "medium-high and low-outside," among which the carbon emissions in areas with better economic foundations are much higher than those in other areas, and the spatial polarization effect is obvious. In addition, there is a significant spatial positive correlation between county carbon emissions. The counties with high-high clusters are relatively stable, mainly concentrated in the Longzhong region, while counties with low-low clusters are slightly reduced, mainly concentrated in the southern ethnic region and the Longdongnan region, and the county carbon emission clusters type has a spatial locking effect. This is mainly due to the large differences in economic scale, industrial structure, and population size in Gansu Province, and the interaction between economic scale and other factors has a more significant impact on the spatial differentiation of carbon emissions. Moreover, the leading influencing factors of county carbon emission differences also have regional differences. Therefore, differentiated and targeted carbon emission reduction strategies need to be implemented urgently. Due to the lack of real county energy consumption statistics, the research results need to be further tested for robustness.

**Keywords:** carbon emissions; spatiotemporal evolution; regional differences; driving factors; Gansu Province

## 1. Introduction

Global warming has become the focus of the international community [1]. In response to climate change, 60% of countries or regions in the world are implementing carbon-neutral strategies in different ways [2–4]. Most countries, represented by the United States, the United Kingdom, the European Union, and Japan, have set a goal of carbon neutrality by 2050, which means the world consensus on carbon emission reduction is gradually forming [5–7]. As the world's largest carbon emitter, China's carbon emissions

reached 10.67 billion tons in 2020, accounting for 30% of the world's total and putting huge pressure on carbon emission reduction. Of course, China has also actively undertaken the responsibility of reducing emissions. At the 75th UN General Assembly, China solemnly pledged to reach an emission peak by 2030 and strive to achieve carbon neutrality by 2060. However, the historical tasks of urbanization and industrialization in China have not yet been completed, the infrastructure urgently needs to be improved, and the proportion of fossil fuel derived energy is still relatively high. The realization of the "dual-carbon" goal and the task of low-carbon and circular development are facing the double test of arduousness and severity. In particular, Gansu Province, an underdeveloped region located in western China, is not only the main supplier of energy resources but also a crucial supporting area of an ecological security barrier. With the proposal of the "Belt and Road" initiative and the ecological protection and high-quality development strategy of the Yellow River basin in recent years, higher requirements have been put forward for the energy efficiency of the local economy. The traditional and extensive high consumption, high input, and high emission development model urgently needs a low-carbon transformation. The county, as the most basic administrative unit in China, accounts for 78% of China's land area, 71.94% of the population, and 51.80% of the GDP, which is the basic spatial unit and carrier of economic development and industrial transfer, and the main battleground for future urbanization development [8]. Compared with provinces and cities, counties can better capture regional heterogeneity, which is of great significance for the adjustment and transformation of economic structure and development models and actively promotes the realization of carbon emission reduction targets [9]. Thus, strengthening research on carbon emissions at the county level is conducive to improving the scientificity, pertinence, and operability of energy conservation and emission reduction measures [10,11].

Recently, relevant scholars have produced many achievements on carbon emissions from different perspectives and geographical scales, mainly focusing on total carbon emissions [12], footprint [13], intensity [14], structure [15], and efficiency [16], the evolution characteristics and action mechanism of the spatiotemporal pattern and its simulation analysis [17–20], the discussion of carbon emission reduction paths and strategies [21,22], and the evaluation of the effectiveness of carbon trading policies [23]. For the accounting of carbon emissions, the academic community not only pays attention to different industries such as agriculture, forestry, industry, transportation, energy, construction, and tourism [24–28] but also continuously emerges carbon research in the fields of housing, service industry, land use, and waste in recent years [29–33], greatly expanding and deepening the research content of carbon emissions. These studies involve global, national, provincial, municipal, county, community, rural, enterprise, etc., including overall studies and case studies. For example, at the provincial and municipal scales, China's carbon emission research shows significant spatial and temporal differentiation and spatial agglomeration characteristics. Regions with high carbon emissions are mainly distributed in North China, Central China, and eastern coastal regions with developed transportation and large energy consumption [34–36]. Provincial-level energy consumption data are easier to obtain so carbon emission research is more concentrated on the provincial and above spatial scales. County-level carbon emissions research has also been increasing in recent years, typically national or county-level research, but different scholars have different accounting methods. For example, Xia et al. and Simayi et al. allocated provincial or municipal energy consumption to each city or county based on the county population and the output value of each industry, respectively, thus obtaining the carbon emissions of each city or county [37,38]. However, the method ignores the differences in energy utilization efficiency between regions, and the accuracy of carbon emission data need to be verified. With the continuous development and application of remote sensing technology, Lyu et al. built a carbon emission estimation model based on night light data and analyzed the temporal and spatial dynamics of carbon emissions from energy consumption in the Yellow River basin in the provinces, cities, counties, and on a grid scale. The results show that the model's goodness of fit and accuracy meet the requirements, and the estimation model is effective, but the

accuracy of the estimation model for refined county-level data still need to be verified [39]. Moreover, the carbon emission pattern is the joint result of the dynamic spatial effects of the county itself and its neighborhood, and the study of carbon emissions in a certain county is not enough to reflect its spatial spillover effect [40]. In addition, increasing carbon emissions is the interaction of GDP growth, population growth, technological change, institutional mechanisms, and energy structure. Based on this, domestic and foreign scholars have used the GTWR model, SDA model, spatial measurement model, and decoupling index to study the driving factors of air pollution and carbon emission [41,42]. The studies show that the increase in carbon emissions is mainly caused by economic development and economic structure [43,44]. At the same time, population agglomeration caused by urbanization also significantly promotes carbon emissions [45]. Capital investment, opening to the outside world, and market mechanisms have also been confirmed to be important factors affecting carbon emissions [46,47]. However, studies also show that the development of science and technology can effectively reduce carbon emissions and improve energy efficiency, which is conducive to the control and reduction in carbon emission intensity in China at the present stage [48].

It is not difficult to find that the current academic circles have produced relatively systematic studies of carbon emissions, which also provide many useful references for this paper. However, most carbon emission studies are based on traditional statistical data that are often limited to the national or provincial level due to data limitations, and it is difficult to refine to the county scale, which cannot provide more strong support for formulating regional and differentiated carbon emission reduction policies; especially in Gansu Province, which is rich in resources and energy, the county is not only an important part of the industrial zone and a contributor to carbon emissions, but also a key administrative unit to implement the goal and policy of "double carbon." However, the research on carbon in the western region is relatively weak. Therefore, it is necessary to study the county carbon emission in Gansu Province, which is located in the underdeveloped western area. At the same time, the current use of night light data to estimate carbon emissions is mainly based on DMSP/OLS data, and the research was primarily conducted before 2013, so it is difficult to dynamically monitor and track the development trend of carbon emissions in recent years. However, since 2012, NPP/VIIRS data have been quite different from the previous data in terms of spatial resolution, pixel brightness value, and other data characteristics, which has become an obstacle and bottleneck for dynamic estimation and monitoring of regional carbon emissions in a long time series. The CEAD night light inversion carbon emission data used in this paper provides a more accurate data basis for district and county-scale research, and its data have continuity and timeliness. In addition, because of differences in urbanization level and economic foundation, the influencing factors of carbon emissions exist variation. Although large-scale research can control the overall situation, it is not conducive to exploring different development stages and regionally targeted differentiated control. Common but differentiated responsibilities for carbon reduction should be reflected across regions. Gansu Province, as an important ecological hub area in the west and the whole country, is driven by rapid industrialization and urbanization, and its carbon emission reduction work is imminent. Based on this, this paper takes the carbon emissions of 87 counties in Gansu Province as the research object, and with the support of ArcGIS spatial analysis function, Gini coefficient, standard deviation ellipse, spatial autocorrelation, and other methods, explores what the temporal and spatial change trend of carbon emissions of counties in Gansu Province? What are the characteristics or laws of evolution? What factors affect its carbon emission level? The purpose is to reveal the difference in the implementation effect of energy conservation and emission reduction measures under different carbon emission levels and provide a theoretical basis for the implementation of phased carbon emission reduction measures.

## 2. Materials and Methods

### 2.1. Study Area

Gansu Province (92°13′~108°46′ E, 32°11′~42°57′ N) is a 42.78 × 10$^4$ km$^2$ narrow and elongated shape region in the upper reaches of the Yellow River and inland of northwest China, including 14 cities (or prefectures) and 87 counties. Based on the natural geographical location and previous studies, Gansu Province can be roughly divided into four parts: the Hexi Corridor region (Jiuquan, Jiayuguan, Zhangye, Jinchang, and Wuwei City), the Longzhong region (Lanzhou, Baiyin, and Dingxi City), the Longdongnan region (Tianshui, Longnan, Pingliang and Qingyang City) and the southern ethnic region (Linxia Prefecture and Gannan Prefecture) (Figure 1). It is located at the intersection of the three natural regions of China's northwest arid region, Qinghai-Tibet alpine region, and eastern monsoon region so the landforms are complex and diverse, and it has important strategic significance in maintaining national ecological security and ecological civilization construction. As an important industrial base in northwest China, Gansu Province achieved rapid development of industrialization and economy under the promotion of the strategy of western development and "strong industrial province," and has now formed a relatively complete industrial system with regional characteristics. However, there is still a considerable gap compared with China and other developed regions. In 2017, the total population of Gansu Province was 26.26 million, and the GDP reached 767.70 billion yuan, accounting for only 1.89% and 0.93% of China, and economic development was relatively slow. The province's secondary industry accounted for 34.34%, and the energy consumption reached 75.38 million tons. At present, it is still transitioning from the early stage of industrialization to the middle stage. Under the interactive coercion of global climate change and human activities, Gansu Province will face the dual pressures of economic growth and greenhouse gases in the future.

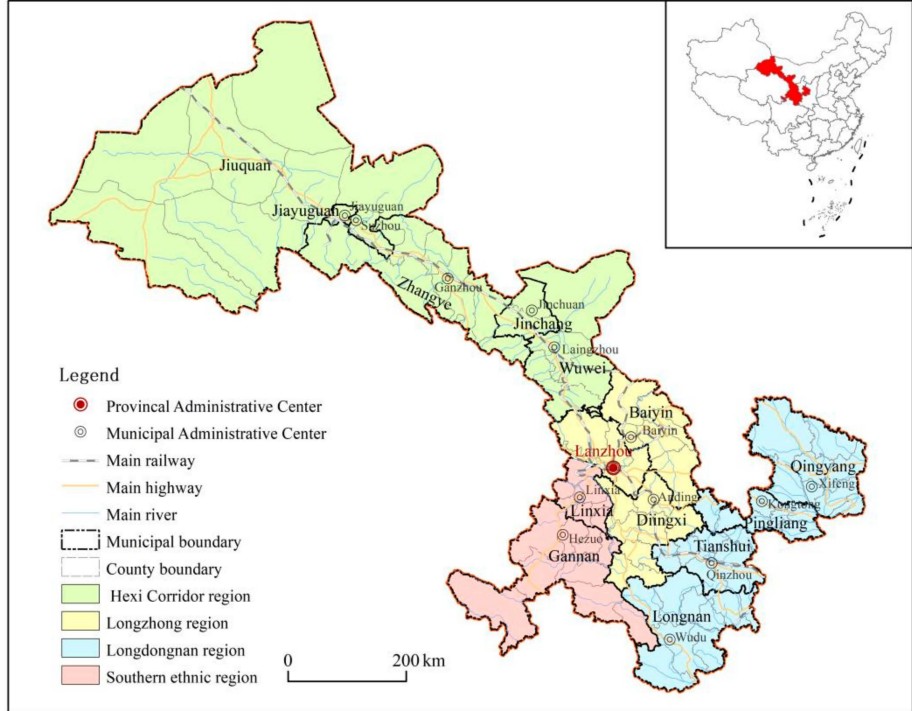

**Figure 1.** Study area.

### 2.2. Data Sources

The county-level carbon emissions data were derived from the CEADs database (http://www.ceads.net (accessed on 18 March 2022)). Since the statistical data on energy consumption are only collected at the provincial-level and cannot be obtained at the county-level, the research on carbon emissions at the county-level needs to be expanded. At

the same time, due to significant differences in satellite data before and after 2013, the accuracy of most carbon emission data estimated and simulated by night light data need to be tested. However, CEADs retrieve county carbon emission data from DMSP/OLS and NPP/VIIRS night light data through particle swarm optimization-back propagation (PSO-BP) algorithm, which has the advantages of consistent statistical caliber, strong continuity, and high accuracy. The socioeconomic data were derived from the Statistical Yearbooks of Gansu.

*2.3. Research Methods*

2.3.1. Gini Coefficient

The Gini coefficient, also known as the Lorenz coefficient, is a commonly used index to measure the income difference among residents in a country or region, which is of great significance for evaluating macroeconomic development and the gap between rich and poor. In recent years, its connotation has been enriched and expanded and has been widely used in resource, environmental, carbon emission, and other fields. Therefore, this paper introduces the environmental Gini coefficient to assess the degree of difference in carbon emission distribution [49]. The calculation formula is as follows:

$$GINI = 1 - \sum_{i=1}^{n} (pop_i - pop_{i-1})(emi_i + emi_{i-1}) \tag{1}$$

where *GINI* is the Gini coefficient of carbon emissions per capita; $pop_i$ is the historical cumulative population proportion of the region; $emi_i$ is the historical cumulative carbon emission proportion of the region. The *GINI* value is [0, 1], and the smaller the value, the smaller the unfairness of regional carbon emissions. According to international practice, 0.4 is generally regarded as the "warning line" of distribution gap or fairness. If *GINI* is less than 0.4, it means that the regional carbon emission allocation is relatively fair. Otherwise, the allocation of carbon emissions is not fair enough.

2.3.2. Standard Deviation Ellipse

The standard deviation ellipse is characterized by centrality, distribution, density, orientation, and morphology, which can accurately describe the directional distribution relationship between carbon emissions and regions and quantitatively describe its dynamic evolution trend [50]. The center of gravity of the ellipse depicts the centrality of the elements (carbon emission), the azimuth depicts the trend of elements, the area depicts the main range of elements distribution, the major axis of the ellipse represents the distribution direction of elements, and the minor axis represents the dispersion of elements. The specific formula refers to Yang et al. [51].

$$\overline{X} = \frac{\sum_{i=1}^{n} x_i}{n}, \; \overline{Y} = \frac{\sum_{i=1}^{n} y_i}{n} \tag{2}$$

$$SDE_x = \sqrt{\frac{\sum_{i=1}^{n} (x_i - \overline{X})^2}{n}}, \; SDE_y = \sqrt{\frac{\sum_{i=1}^{n} (y_i - \overline{Y})^2}{n}} \tag{3}$$

$$\tan\theta = \frac{\sum_{i=1}^{n} \widetilde{x}_i^2 - \sum_{i=1}^{n} \widetilde{y}_i^2 + \sqrt{\left(\sum_{i=1}^{n} \widetilde{x}_i^2 - \sum_{i=1}^{n} \widetilde{y}_i^2\right)^2 + 4\left(\sum_{i=1}^{n} \widetilde{x}_i \widetilde{y}_i\right)^2}}{2\sum_{i=1}^{n} \widetilde{x}_i \widetilde{y}_i} \tag{4}$$

In the formula, $\overline{X}$ and $\overline{Y}$ are the central coordinates of the ellipse, respectively; $x_i$ and $y_i$ are the coordinates of the *i*-th study unit, respectively; $SDE_x$ and $SDE_y$ are the variances of the *x*-axes and *y*-axes of the ellipse, respectively; $\tan\theta$ is the rotation angle of the ellipse; $\widetilde{x}_i$

and $\widetilde{y}_i$ are the deviation between the center of the ellipse and the center of the *i*-th element space, respectively.

### 2.3.3. Spatial Autocorrelation Analysis

The first law of geography states that geographical phenomena are spatially correlated. Spatial autocorrelation reflects the correlation degree between a certain attribute value on a spatial unit and the same attribute value on its neighbors. To measure the overall characteristics of the spatial correlation degree of carbon emissions, the global Moran's *I* statistic was used [52]. The equation is as follows:

$$I = \frac{n \sum\limits_{i=1}^{n} \sum\limits_{j=1}^{n} w_{ij}(x_i - \overline{x})(x_j - \overline{x})}{\sum\limits_{i=1}^{n} \sum\limits_{j=1}^{n} w_{ij} \sum\limits_{i=1}^{n} w_{ij}(x_i - \overline{x})^2} \tag{5}$$

where $x_i$ and $x_j$ represent the observed values of unit *i* and unit *j*, respectively; *n* is the number of space units; $\overline{x}$ is the average of the observations; $w_{ij}$ is the spatial weight. $I \in [-1, 1]$. When $I > 0$, the observed values present spatial positive autocorrelation; When $I < 0$, the observed values present negative spatial autocorrelation. The absolute value of the *I* index reflects the degree of spatial correlation. The larger the absolute value, the larger the degree of spatial correlation.

To reveal the local similarities and differences in the spatial association of carbon emissions between local units and adjacent units, this paper further conducts clustering tests through local spatial autocorrelation (*LISA*) [53]. The calculation formula is:

$$I_i = \frac{n(x_i - \overline{x}) \sum\limits_{j=1}^{n} w_{ij}(x_j - \overline{x})}{\sum\limits_{i=1}^{n} (x_i - \overline{x})^2} \tag{6}$$

In the formula, if $I_i > 0$, it indicates that the carbon emission distribution trend between adjacent units is similar, which is a high-high or low-low value cluster; If $I_i < 0$, it indicates that the distribution trend of carbon emissions between adjacent units is not similar, which is a high-low or low-high value clusters.

### 2.3.4. Geodetector Model

Referring to the existing research, following the principles of availability of county data and scientific and typical construction of indicator systems, this paper focuses on the investigation and in-depth analysis of the impact of 9 factors, including population, economy, investment, and technology, on carbon emissions (Table 1). Studies have confirmed that population is one of the main driving factors of carbon emissions [54–56], and population has an impact on carbon emissions through production and consumption behavior and has a two-way effect. Compared with population density, structure, and other indicators, the population size and urbanization level selected in this paper can better reflect the direct impact of population changes on carbon emissions in the process of urbanization. The promotion of the regional economy makes changes in production and consumption patterns and then changes the level of carbon emissions. Therefore, this paper selects economic scale and financial input to characterize the level of affluence that affects environmental pressure because the irrational industrial structure is the contributing factor to China's carbon emissions, and the economy of Gansu Province is generally underdeveloped, most counties are still in the middle stage of industrialization development, so the ratio of the added value of secondary production to the added value of primary production is used to discuss the impact of industrial structure on county carbon emissions. Consumption is an important carriage for urban economic development, and the improvement of the county's disposable income will promote an increase in urban

consumption emissions [57]. Therefore, this paper selects the per capita disposable income of urban residents to represent the impact of living standards on carbon emissions. The market potential of large-scale and high-consumption activity attracts the migration of manufacturing and service industries, which will have an impact on carbon emissions. Therefore, this paper selects market scale and investment level to characterize the impact on carbon emissions. Industrialization plays a decisive role in reducing carbon emissions, and the improvement of energy efficiency is one of the key means to achieve carbon emission reduction [58]. This paper selects carbon emission intensity to characterize the impact of technological progress on carbon emissions. First, SPSS 27 software was used to conduct KMO and Bartleet tests for the driving factors, and it was calculated that the KMO value was greater than 0.5, and the Bartleet spherical test results were all significant at the 1% level, indicating that the variables could be factor analyzed.

**Table 1.** The main explanatory variables.

| Influence Factors | Index Definition | Mean Value | Standard Deviation | Minimum Value | Maximum Value |
|---|---|---|---|---|---|
| Population size ($X_1$) | Total resident population at the end of the year (10,000 persons) | 30.34 | 20.31 | 1.53 | 131.61 |
| Urbanization level ($X_2$) | Proportion of urban population in total resident population (%) | 37.68 | 20.99 | 11.04 | 100.00 |
| Economic scale ($X_3$) | Gross domestic product (10,000 yuan) | 83.36 | 121.34 | 7.78 | 947.13 |
| Industrial structure ($X_4$) | Ratio of secondary production value added to primary production value added | 12.17 | 49.01 | 0.22 | 358.60 |
| Financial input ($X_5$) | General public budget expenditure (10,000 yuan) | 25.06 | 9.70 | 7.44 | 63.68 |
| living standard ($X_6$) | Per capita disposable income of urban residents (10,000 yuan) | 2.46 | 0.48 | 1.67 | 3.87 |
| market size ($X_7$) | Per capita retail sales of consumer goods (10,000 yuan) | 1.10 | 0.99 | 0.10 | 5.54 |
| Investment level ($X_8$) | Investment in fixed assets per capita (10,000 yuan) | 2.53 | 2.38 | 0.38 | 20.14 |
| Carbon emission intensity ($X_9$) | Carbon emission per unit GDP (t/10,000 yuan) | 3.03 | 1.47 | 0.65 | 8.63 |

Geographic detectors have been widely used in natural, social economic, environmental, and other related fields in recent years because of their few assumptions and wide applications. Therefore, this paper uses actor detection and interaction detection in geographic detectors to detect the leading factors affecting the spatial differentiation of carbon emissions in counties of Gansu Province [59,60]. Its calculation formula is:

$$q = 1 - \frac{\sum\limits_{h=1}^{L} n_h \sigma_h^2}{n\sigma^2} \tag{7}$$

where $q$ is the explanatory power value of the probe factor; $L$ is each factor type; $n$ and $n_h$ are the number of all samples in the study area and the number of samples within the factor type $h$, respectively; $\sigma^2$ is the discrete variance of all samples in the whole study area; $\sigma_h^2$ is the variance of sample discrete variance within factor type $h$; $q \in [0, 1]$. The larger the $q$ value, the stronger the spatial determinacy of the factor to carbon emissions.

In addition, the interaction detection model is used to detect whether each driving factor has an independent influence when explaining factor variables or whether it has enhanced or weakened interpretation ability after an interaction. The interaction types of factors mainly include five types: nonlinear attenuation, single-factor nonlinear attenuation, double-factor enhancement, and independent and nonlinear enhancement.

## 3. Results and Analysis

### 3.1. Temporal Evolution of Carbon Emissions

The county total carbon emissions in Gansu Province showed an upward trend of "first urgent and then slow" from 1997 to 2017 (Figure 2a). Although carbon emissions fluctuated up and down after 2011, the decline was small, indicating that Gansu Province

still faces severe pressure to reduce emissions. The per capita carbon emissions in the counties of Gansu Province increased from 2.17t in 1997 to 6.53t in 2017, an increase of 200.92%, especially before 2011. After 2011, the increase rate of per capita carbon emissions slowed down significantly and reduced in 2015, which is inseparable from the policy constraints and implementation of high-quality economic transformation since the 18th National Congress and energy conservation and emission reduction targets. From the regional perspective, the carbon emissions in the Longzhong region have always been the largest, reaching 65.48 million tons in 2017 and accounting for 38.21% of Gansu Province. The carbon emissions in the Longdongnan region lagged behind the Hexi region before 2005 and rose rapidly after 2005, overtaking the Hexi region and becoming the second largest carbon emission area in Gansu Province, which benefited from the optimization of industrial structure and the effective promotion of ecological protection and restoration projects in Hexi region. The carbon emissions of the southern ethnic minority region are always lower than other regions.

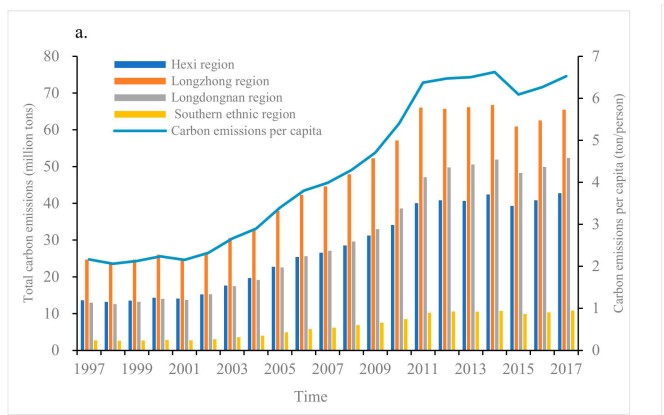
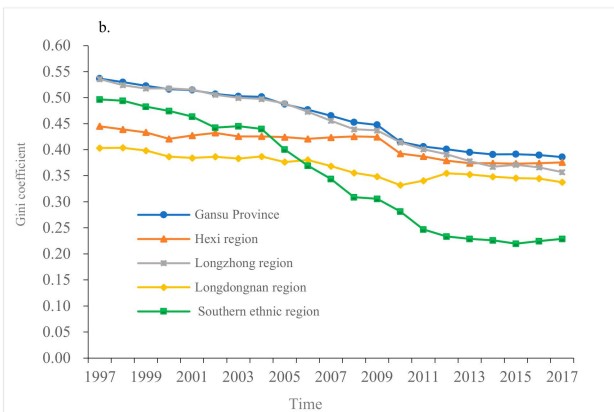

**Figure 2.** Regional differences of carbon emissions and changes of Gini coefficient of Gansu Province in 1997–2017. (a. Carbon emissions in different regions; b. Gini coefficient of carbon emissions.)

The Gini coefficient of county carbon emissions in Gansu Province showed a continuous downward trend from 1997 to 2017 (Figure 2b), decreasing from 0.54 in 1997 to 0.39 in 2017, indicating that the carbon emission gap between counties was narrowing. The Gini coefficient of county carbon emissions in the Hexi region generally shows a fluctuating downward trend, and after 2009, the Gini coefficient fell below the warning value of 0.4, and the carbon emission gap decreased slightly. The change in the Gini coefficient of county carbon emissions in the Longzhong region is basically consistent with that in Gansu Province, and the decline is slightly faster than that in Gansu Province after 2012, but the average value of the Gini coefficient is still above 0.4. The Gini coefficient of carbon emissions in the Longdongnan region shows a fluctuating downward trend, and the Gini coefficient is not significantly higher than 0.4 during the study period. Hence, the difference in county carbon emissions is relatively fair. The Gini coefficient of carbon emissions of the southern ethnic region has the largest downward trend, especially after 2006, and the difference in county carbon emissions is significantly smaller than that in other regions.

*3.2. Spatial Pattern of Carbon Emissions*

The carbon emission levels in the counties of Gansu Province showed an obvious upward trend from 1997 to 2007. In 1997, county carbon emissions of Gansu Province were relatively low as a whole. Except for cities and industrial cities, the carbon emissions of other counties did not exceed one million tons. In 2007, carbon emissions of all counties increased, especially in the Longzhong region, with Lanzhou as the core. By 2017, the increase in carbon emissions in the counties had obviously accelerated and gradually spread to the periphery, forming a geospatial pattern of medium-high and low outside (Figure 3). The distribution of areas with higher carbon emissions and above in counties is scattered,

mainly concentrated in the Longzhong region represented by Yuzhong, Chengguan, Qilihe, Honggu, Yongdeng, and Baiyin District, and the regional urban areas in the Hexi region. The low-value areas of carbon emissions are concentrated in the southern ethnic region, mainly in Gannan Prefecture, and the carbon emissions in most counties are lower than one million tons.

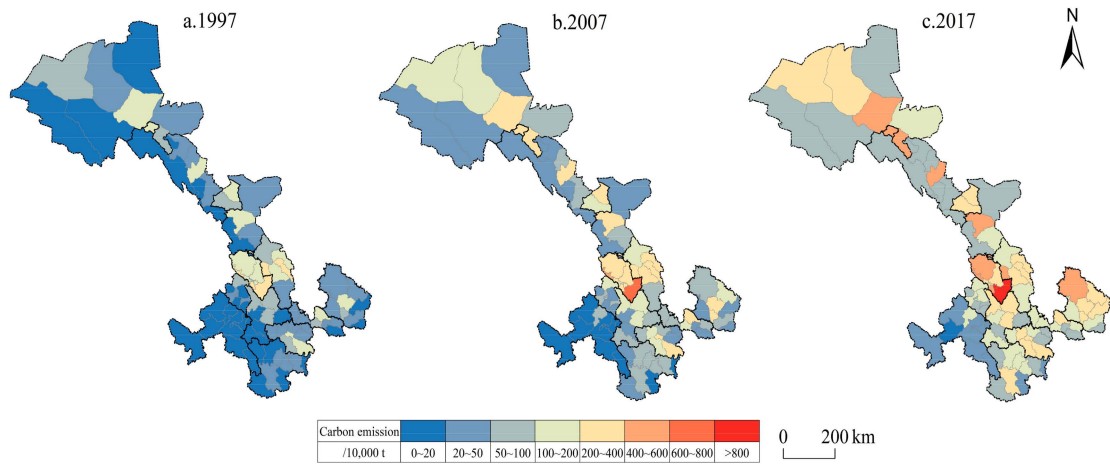

**Figure 3.** Spatial patterns of county carbon emissions of Gansu Province in 1997–2017.

### 3.3. Evolution Trajectory Analysis of Carbon Emissions

In this part, five-time points in 1997, 2002, 2007, 2012, and 2017 were selected to measure the spatial movement characteristics of county carbon emissions of Gansu Province and the relevant attributes of the standard deviation ellipse (Figure 4).

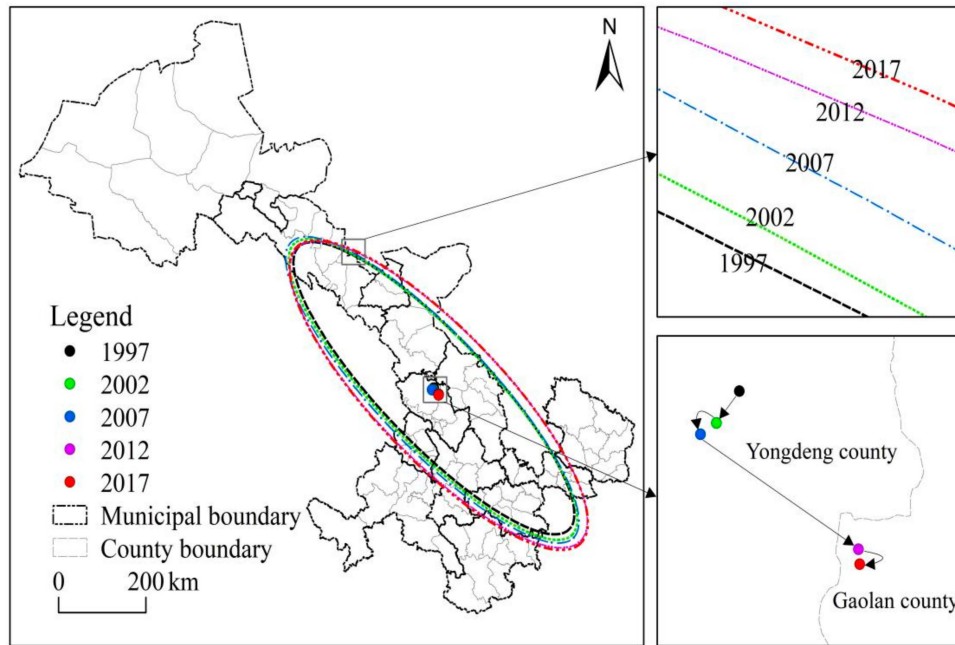

**Figure 4.** The standard deviation ellipse and movement trajectory of gravity center of county carbon emissions of Gansu Province in 1997–2017.

The distribution pattern of county carbon emissions in Gansu Province is basically stable, and the carbon emissions center is always located in Lanzhou City in the Longzhong region, manifesting that carbon emissions at the county level of Lanzhou City are relatively high, which is basically consistent with the characteristics of its spatial evolution pattern. From 1997 to 2002 and from 2002 to 2007, the center of carbon emission moved to the

southwest two consecutive times. From 2007 to 2012, the center of carbon emission shifted significantly to the southeast, and from 2012 to 2017, it shifted slightly to the south. In terms of moving distance, the center of county carbon emissions distribution of Gansu Province moved to the southeast more than the total distance to the southwest and south, indicating that the increasing rate of county carbon emission in the Longdongnan region was fast after 2007, and driving the center of carbon emission in Gansu Province to shift to the southeast gradually.

From the perspective of long and short semi-axes, the long semi-axis showed a fluctuating elongation trend in the northwest-southeast direction from 1997 to 2017, with the radius increasing from 428.56 km to 449.83 km, and the short semi-axis also showed an elongation trend in the northeast-southwest direction, with the radius increasing from 117.05 km to 140.23 km, indicating that there was a spatial diffusion trend of carbon emissions in the northwest-southeast and northeast-southwest directions. From the azimuth angle, the azimuth angle of carbon emissions decreased slightly from 1997 to 2017, indicating that the divergence direction of carbon emissions was relatively stable. From the elliptical area, the elliptical area of carbon emissions has expanded from 157,548 $km^2$ in 1997 to 198,135 $km^2$ in 2017, illustrating the high-quality development of Gansu's social economy is still facing greater carbon emission pressure.

*3.4. Spatial Agglomeration Analysis of Carbon Emissions*

The Moran's *I* values of county carbon emissions in Gansu Province were 0.349, 0.319, and 0.275 in 1997, 2007, and 2017, respectively, and the Z-value of normal statistics passed the 1% significance test, that is, county carbon emissions in Gansu Province showed a positive spatial autocorrelation, which indicates that there are high (low) value clusters of county carbon emissions in space, and the spatial agglomeration degree of county units with similar carbon emissions decreases.

Figure 5 shows that the spatial agglomeration pattern of county carbon emissions of Gansu Province was relatively fixed from 1997 to 2017, dominated by high-high clusters and low-low clusters. High-high clusters were basically unchanged, mainly distributed in Lanzhou, Baiyin, and other Longzhong regions. Low-low clusters were mainly distributed in Gannan, Longnan, Linxia, and Dingxi counties in the southwest of Gansu Province, and the range of low-low clusters was slightly reduced.

To further reveal the transfer of county carbon emissions in different local relevant spatial patterns at different times, the LISA time-space transition matrix is further used for tracking analysis. Spatiotemporal transitions can be divided into four types, involving 16 transition forms. Type I refers to the transition that only occurred in the county itself; Type II refers to those that the county itself has not transited, and the neighborhood shape has transited; Type III refers to the transition of the county itself and its neighborhood; Type IV refers to the fact that neither the county itself nor its neighborhood has experienced any transition. Table 2 shows that the type IV transition was the main type, and few transitions occurred among different types. There was a certain transition inertia among county types, and the evolution of carbon emissions was "path dependent." Among them, the spatiotemporal cohesion index of type IV is as high as 0.88, which means that the probability of no spatiotemporal transition of carbon emissions is 88%, demonstrating that the spatial cohesion of carbon emissions at the county of Gansu Province shows a high path locking effect.

**Table 2.** LISA spatiotemporal transition matrix of carbon emissions of Gansu Province in 1997–2017.

| Time | Spatial Association Mode | $HH_{t+1}$ | $HL_{t+1}$ | $LL_{t+1}$ | $LH_{t+1}$ |
|------|--------------------------|------------|------------|------------|------------|
| 1997–2017 | $HH_t$ | IV(12) | II(0) | III(0) | I(1) |
| | $HL_t$ | II(0) | IV(1) | I(0) | III(0) |
| | $LL_t$ | III(0) | I(1) | IV(13) | II(0) |
| | $LH_t$ | I(1) | III(0) | II(0) | IV(3) |

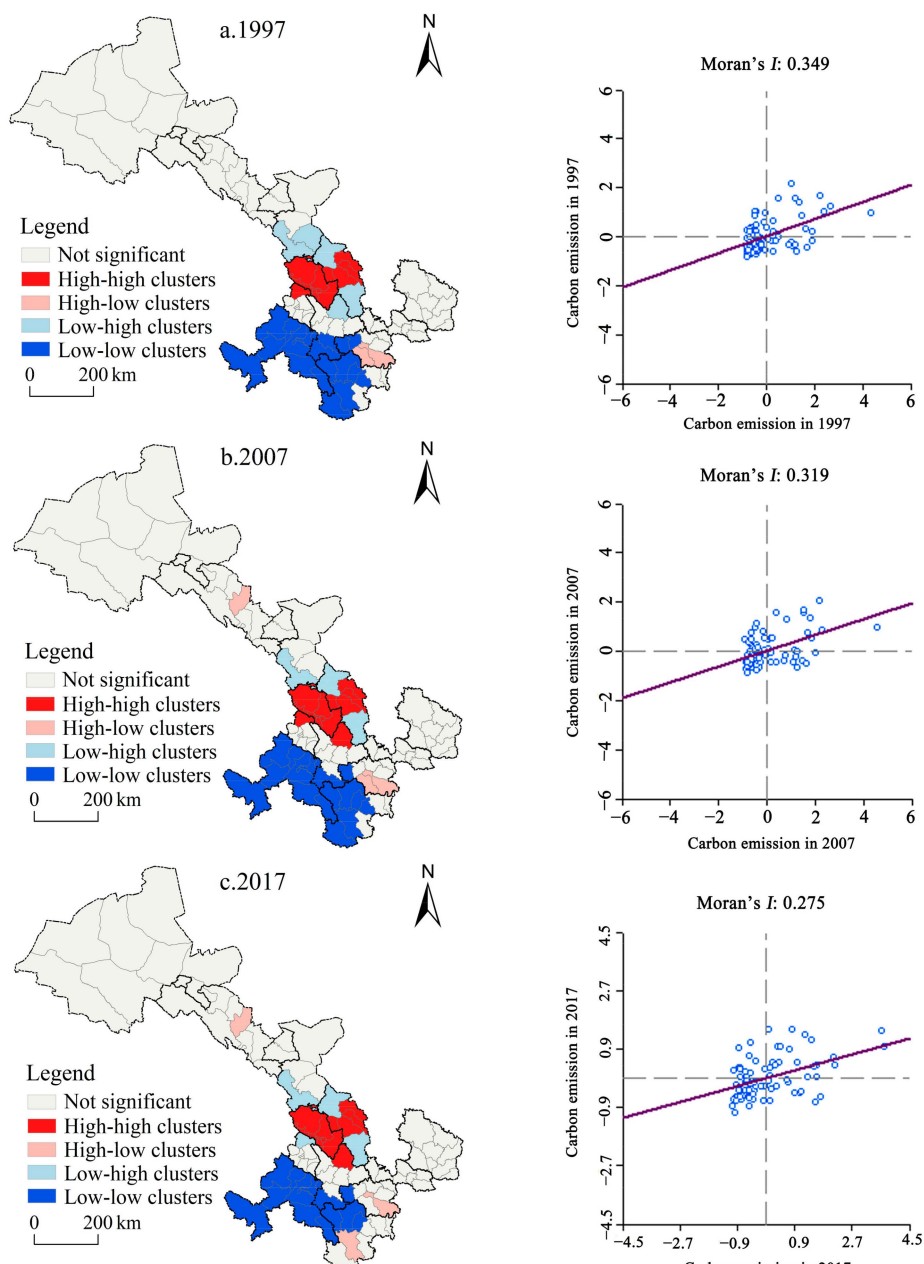

**Figure 5.** Moran scatter plot and LISA clustering of carbon emissions of Gansu Province in 1997–2017.

## 4. Driving Factors of Carbon Emissions

*4.1. Driving Factors of Spatial Differentiation of Carbon Emissions in Gansu Province*

The spatial differentiation of county carbon emissions of Gansu Province is mainly mutually affected by economic, societal, population, and other factors. As shown in Table 3, economic scale and industrial structure are the main factors for the increase in carbon emissions, with *q* values reaching 0.760 and 0.531, respectively. Meanwhile, the *q* value of population size and urbanization level is also higher than 0.440, indicating that urbanization level and population size are also important factors affecting carbon emissions. Urbanization mainly leads to an increase in energy consumption and carbon emissions through the population scale effect, urban scale expansion, and industrial agglomeration. In addition, the impact of living standards on carbon emissions cannot be ignored either.

The interaction detection results show that the dominant interaction factors have stronger explanatory power than the single factor, and the interaction between factors is

mainly nonlinear and enhanced, which indicates the spatial variation of carbon emissions in Gansu Province is mainly the comprehensive effect of multiple factors (Figure 6a). Among them, the explanatory power of interaction between economic scale and other factors is significantly higher than that of interaction between other factors. The explanatory power of interaction between economic scale and carbon emission intensity is the strongest, the $q$ value reaches 0.996, and the explanatory power of interaction between financial input and investment level is the weakest. For Gansu Province, taking advantage of the opportunity of the new round of western development, reducing carbon emissions through technological innovation, industrial structure transformation, upgrading, improving energy efficiency, and other two-pronged measures is the key to achieving low-carbon development.

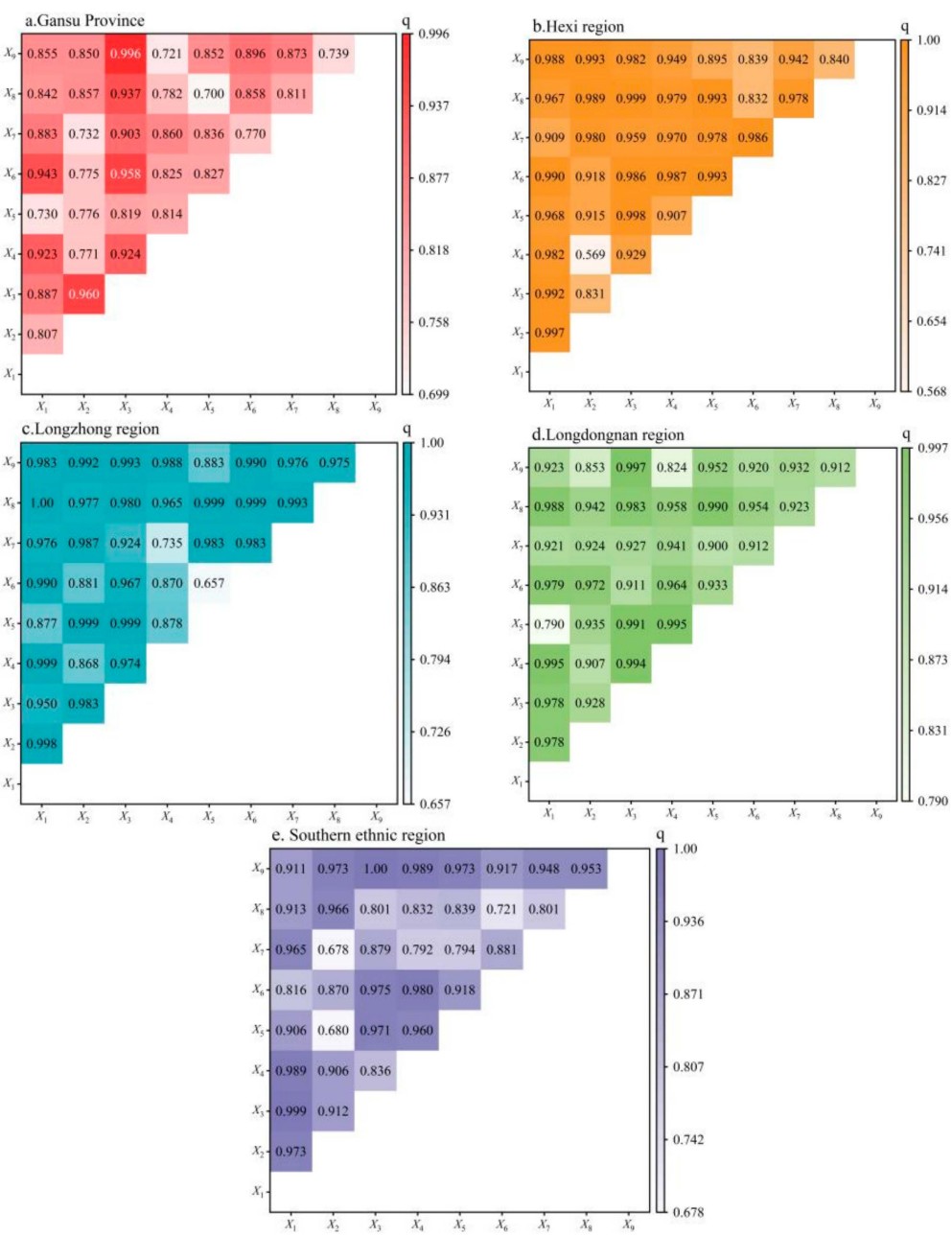

**Figure 6.** Interaction detection results of carbon emission driving factors in Gansu province and sub-regions.

**Table 3.** Detection results of the driving factors of carbon emissions in Gansu Province and its sub-regions.

| Factors | $q$ | | | | |
|---|---|---|---|---|---|
| | **Gansu Province** | **Hexi Region** | **Longzhong Region** | **Longdongnan Region** | **Southern Ethnic Region** |
| $X_1$ | 0.445 *** | 0.645 *** | 0.482 *** | 0.547 *** | 0.816 *** |
| $X_2$ | 0.414 *** | 0.259 *** | 0.475 *** | 20.475 *** | 0.340 *** |
| $X_3$ | 0.760 *** | 0.804 *** | 0.768 *** | 0.797 *** | 0.764 *** |
| $X_4$ | 0.531 *** | 0.456 *** | 0.630 *** | 0.670 *** | 0.580 *** |
| $X_5$ | 0.233 *** | 0.664 *** | 0.204 *** | 0.507 *** | 0.478 *** |
| $X_6$ | 0.441 *** | 0.726 *** | 0.449 *** | 0.686 *** | 0.667 *** |
| $X_7$ | 0.333 *** | 0.855 *** | 0.458 *** | 0.615 *** | 0.590 *** |
| $X_8$ | 0.334 *** | 0.646 *** | 0.707 *** | 0.154 *** | 0.410 *** |
| $X_9$ | 0.299 *** | 0.718 *** | 0.706 *** | 0.406 *** | 0.844 *** |

Notes: *** $p < 0.01$.

*4.2. Driving Factors of Carbon Emissions in Different Regions*

Table 3 and Figure 6b–e show that the influencing factors of spatial differentiation of county carbon emissions in different regions are heterogeneous.

The leading factors of carbon emission in the Hexi region are market size (0.855), economic size (0.804), and economic size ∩ investment level (0.999). The Hexi Corridor is an important military town in the northwest of China and an indispensable part of the ancient Silk Road. It is rich in resources and energy. In its early days, it was a rich place for military strategists to compete. The urban function of the "tea and horse trade" has stimulated the expansion of trade and consumption markets in this region and the constant increase in social fixed investment. In addition, the rapid development of resource-based cities such as Yumen, Jinchuan, Jiayuguan, and Suzhou has injected vitality into the economy. However, the rapid development and extensive utilization of resources and energy will inevitably promote an increase in carbon emissions. With the depletion of resources in recent years, its industry is facing the challenge of adjustment and transformation, and it is urgent to promote the development of the local economy to a green environment.

The leading factors of carbon emission in the Longzhong region are economic size (0.768), investment level (0.707), and population size ∩ investment level (0.999). Lanzhou and Baiyin, as important industrial towns in the west, economic centers in the upper reaches of the Yellow River, and important cities in the Silk Road Economic Belt, have a fast flow of regional factor resources and strong regional radiation. The economic foundation of the counties in this region is better than that of other regions as a whole. With the further promotion of the construction of Lanzhou-Xining urban agglomeration, the market radiation scope is gradually expanding, production factors are concentrated, population concentration is high, and the scale of construction land is expanding. However, due to the terrain limitation of the valley basin, the dense population and production activities have increased the atmospheric environmental pressure, making it a high-value concentration area of carbon emissions. In addition, heavy industries with high water consumption, high energy consumption, and high pollution, such as the petrochemical and nonferrous metal smelting industry, still account for a large proportion of the region's industries. The convergence of economic structure is common, and it is difficult to change the transformation and upgrading of the economic industrial structure and the conversion of new and old kinetic energy in the short term. While the high-energy-consuming industries promote economic development, they also contribute to the increase in carbon emissions in the region.

The dominant factors of carbon emission in the Longdongnan region are economic size (0.797), living standard (0.686), and economic size ∩ carbon emission intensity (0.997). The region has laid a solid foundation for the rapid promotion of the county economy by virtue of its favorable location advantages and good industrial foundation such as electronics, electricians, and equipment manufacturing. However, traditional industries still account

for a certain proportion, and green manufacturing with technological innovation as the core is relatively limited. With the improvement of the living standard of urban residents, the concept of urban consumption has gradually changed, and the consumption of various goods and services has increased. The potential of large-scale and high-consumption markets has attracted the migration of manufacturing and service industries, and the living standard has increased carbon emissions. In addition, the region has a large national energy and chemical industry base. Pingliang and Qingyang have achieved rapid economic growth by virtue of their energy endowment advantages and have long been at a higher carbon emission level in Gansu Province. However, the energy structure dominated by coal has promoted the agglomeration of high energy-consuming industries, which is not conducive to the realization of the "low carbon" and "green" development goals in the region.

The main influencing factors of carbon emission in the southern minority region are carbon emission intensity (0.844), population size (0.816), and economic size ∩ carbon emission intensity (1.00). This region is an arid and alpine agricultural and pastoral area at the intersection of Gansu and Qinghai. The ecosystem is fragile, the industrial foundation is weak, and investment is limited. Agriculture and animal husbandry are the main channels of the county economy. A higher livestock breeding scale, while increasing income, also leads to increased carbon emissions. At the same time, restrictions emplaced by the natural environment, economic foundation, traditional ways of production and living, and other energy inputs. This region has an extensive household energy consumption structure and low energy utilization efficiency. The increase in population and the development of urbanization promote the increasing energy consumption of buildings and living, which leads to the increase in carbon emissions from production and living.

## 5. Discussion

Reducing carbon emissions has become a global goal in combating climate change. According to the spatiotemporal evolution trajectory and driving factor analysis of county carbon emissions, Gansu Province should note the following points in terms of carbon peak and emission reduction path:

The above analysis results show that Gansu Province is still facing a severe situation in carbon emissions in the future. Therefore, Gansu Province should fully grasp the strategic opportunities of "double carbon," "the Belt and Road," the new round of western development, ecological protection, and high-quality development of the Yellow River basin, and combine the background foundation and resource endowment conditions of each region to strengthen the investment in low-carbon technology innovation, improve the policies and development mechanisms related to carbon emissions, give play to the ability of low-carbon technology to optimize and upgrade traditional industries, accelerate the cultivation and expansion of distinctive industries, encourage the integration of upstream and downstream development of different industries and industrial chains, and avoid homogeneous competition between counties, enterprises, and industries. To effectively learn from the experience of advanced low-carbon pilot cities, scientifically formulate systematic and diverse green low-carbon development models and emission reduction paths based on local conditions, and strive to achieve a peak in carbon emissions before 2030.

Unbalanced and inadequate development is still the major contradiction of carbon emissions in Gansu Province. The urban areas of prefecture-level cities and industrial-oriented districts (cities and counties) are the basic implementation units of carbon emission reduction in the future. As the largest source of carbon emissions in Gansu Province, the Longzhong region is a key area for carbon emission reduction at present. Meanwhile, the Longzhong region is also the core region of the province. In the future, Longzhong should play a leading role in carbon emission reduction, strive to be a proponent and practitioner of promoting carbon peak and carbon neutralization, take carbon reduction as the key strategic direction, promote the upgrading of traditional industries such as the petrochemical and non-ferrous metal smelting industries, and eliminate production capacity with high emissions, high consumption, and low efficiency, continuously improve

the energy consumption structure of production and living. Moreover, the region should improve the mechanisms of carbon emission reduction, carbon trading, and energy trading in different industries in an orderly manner, strengthen the investment and application of green low-carbon technologies, continuously optimize energy resource allocation capabilities, expand the clean energy consumption and investment market, and strive to become a demonstration area for energy conservation and emission reduction in the province.

Economic scale is the main driving force for the increase in county carbon emission in Gansu Province, but it is obviously inconsistent with the actual economic development to achieve carbon emission reduction by abandoning economic growth and limiting industry scale output. Therefore, under the premise of maintaining steady economic growth, Gansu's carbon emission reduction policy needs to promote the integration and development of energy resources advantages and industrial cultivation so as to realize the coordinated development of carbon emission reduction and economy and society. For the Longzhong region where energy consumption is relatively concentrated, give full play to the advantages of human capital, economic capital and technological innovation and research and development, promote the deep integration of energy, science and technology, economy and industry, and form a leading role in the demonstration of leading industries. For the Hexi region, take full strength of resource endowment, vigorously develop wind power, photovoltaic power generation, solar heat and other clean new energy, and focus on key technologies of new energy, strengthen coordination and linkage of production, education and research, and build a complete industrial chain. For the Longdongnan region, constantly consolidate the economic foundation, adhere to the pace of green and clean development and supply of coal, oil, natural gas and other resources, continuously optimize the energy structure and layout, strengthen technological innovation capabilities. While maintaining economic growth, the southern ethnic region should grasp the ecological protection and high-quality development strategy of the Yellow River Basin, adhere to ecological priority, actively promote new energy sources of wind, solar and electricity, and improve the energy utilization efficiency.

## 6. Conclusions

From the multi-dimensional perspective of time and space, this paper analyzed the overall change trend, regional differences, and spatial pattern of carbon emissions in 87 countries of Gansu Province by Gini coefficient, ArcGIS spatial analysis function, standard deviation ellipse, and spatial autocorrelation. At the same time, the driving factors of carbon emissions were detected by geographic detectors. The purpose of this study is to provide a reference for the realization of carbon emission reduction targets in Gansu Province in order to promote the green, low-carbon, and coordinated development of Gansu Province. The results show that:

County carbon emissions in Gansu Province showed an upward state of "first urgent and then slow" from 1997 to 2017, with an overall convergence tendency, but it has not yet reached the carbon peak. From the perspective of regional differences, the Longzhong region is the main source of carbon emissions in Gansu Province, and the differences within counties are still large. The carbon emissions in the southern ethnic region are always the smallest, and the differences in carbon emissions between counties are also small.

In terms of spatial distribution, county carbon emissions in Gansu Province show a spatial pattern of "high in the middle and low at both ends." The carbon emissions in the Longzhong region are significantly higher than other regions, and provincial capitals, industrial cities, and prefecture-level cities are much higher than general county units and with the passage of time, the center of gravity of carbon emission distribution moved slightly to the southeast direction, but the overall change was little. Meanwhile, county carbon emissions have an obvious spatial positive correlation. The distribution range of the high-high clusters area is relatively stable, mainly concentrated in Lanzhou City and Baiyin City in the Longzhong region, while the low-low cluster areas are widely distributed, mainly in the southern ethnic region and Longnan City in the Longdongnan region. In

addition, the LISA spatiotemporal transition matrix shows that there is a high path-locking effect in county carbon emissions.

From the analysis of driving factors, we can see that economic scale, industrial structure, and population size are the dominant factors for the spatial differentiation of county carbon emissions in Gansu Province, among which economic scale has the strongest explanatory power and the explanatory power is significantly enhanced after interaction with other factors. The leading factors of the spatial differentiation of carbon emission levels in different regions are quite different. The market scale and carbon emission intensity have a significant impact on the Hexi region and the southern ethnic region, while the economic scale has a significant impact on the Longzhong region and Longdongnan region.

The carbon emission data retrieved by CEADs night lights used in this paper provide a more accurate data basis for county-level research, fill the gap of statistical data in micro-scale energy data, and are widely used by domestic and foreign scholars in the assessment of socioeconomic factors such as population, urban expansion and traffic, and the measurement and research of pollutants and carbon emissions. Based on these data, this paper also explores the spatiotemporal evolution characteristics of carbon emissions at the county level in Gansu Province in order to provide a reference for the development of carbon emission reduction strategies in different regions. However, due to the lack of real county-level energy consumption statistical data and the latest county carbon emission data, some studies cannot be carried out, or the results of the studies vary greatly or are not accurate enough. Therefore, it is recommended that the county government of Gansu Province establish an energy consumption statistics system, develop an energy balance sheet, and directly obtain carbon emission data, which provides a basis for formulating more effective carbon emission reduction policies in the county. In the future, in addition to night light data, multi-dimensional data such as the land, social economy, and natural geography should be considered when calculating carbon emissions, which can not only expand the application scope of the night light data set, provide a reference for the government and policymakers, but also help to achieve coordinated emission reduction between different departments.

**Author Contributions:** W.Z. designed the study and processed the data; W.H. and P.S. commented on the manuscript; All authors formed a significant contribution to the results, related discussions, and manuscript writing. All authors have read and agreed to the published version of the manuscript.

**Funding:** This research was funded by the National Natural Science Foundation of China (Grant No. 41771130, 41661035, 42161043) and the Gansu Provincial Science and Technology Program (20JR5RA529).

**Institutional Review Board Statement:** Not applicable.

**Informed Consent Statement:** Not applicable.

**Data Availability Statement:** The data presented in this study are available on request from the first author.

**Acknowledgments:** The authors would like to thank the editors and reviewers for their suggestions and comments.

**Conflicts of Interest:** The authors declare no conflict of interest.

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
