# Peer review of "Research on Spatial and Temporal Pattern Evolution and Driving Factors of County Carbon Emissions in Underdeveloped Regions: Gansu Province of Western China as an Example"

_sustainability, doi:10.3390/su15010291_

Round 1
Reviewer 1 Report
This paper studied the overall changes, regional differences, spatio-temporal variation and clustering characteristics of carbon emissions of 87 counties in Gansu Province. the research content is interesting. The results can provide some reference for carbon emission reduction and policy formulation in Gansu Province.
Please provide data on carbon emissions in 87 counties. Please refine the conclusions, divided into research findings and policy recommendations.
Author Response
Response: Thanks for the reviewers for their confirmation and comments. Based on your opinion, we have answered your questions one by one. The revised part of the text has been marked in blue font.
Firstly, the county carbon emission data in this paper mainly comes from China Carbon Accounting Database (CEAD) (https://www.ceads.net/data/county/), which is retrieved from DMSP/OLS and NPP/VIIRS night light data provided by National Physical Geodata Center (NGDC). The data have the advantages of consistent statistical caliber and strong continuity. In the data source part of the article, the corresponding explanation is also made. If reviewers and other need, can follow the data source section of the link to download.
Secondly, for the conclusion part, we further condensed and deepened. On the basis of reference to the submission requirements, article distribution structure, specific format and previous references of the journal, we have put the policy recommendations in the discussion section and are looking forward to your further review. The conclusions in the text are modified as follows:
From the multi-dimensional perspective of time and space, this paper analyzed the overall change trend, regional differences and spatial pattern of carbon emissions in 87 countries of Gansu Province by Gini coefficient, ArcGIS spatial analysis function, standard deviation ellipse and spatial autocorrelation. At the same time, the driving factors of carbon emissions were detected by geographic detectors. The purpose of this study is to provide reference for the realization of carbon emission reduction targets in Gansu Province, in order to promote the green, low-carbon and coordinated development of Gansu Province. The results show that:
County carbon emissions in Gansu Province showed an upward state of "first urgent and then slow" from 1997 to 2017, with an overall convergence tendency, but it has not yet reached the carbon peak. From the perspective of regional differences, Longzhong region is the main source of carbon emissions in Gansu Province, and the differences within counties are still large. The carbon emissions in southern ethnic region are always the smallest, and the differences in carbon emissions between counties are also small.
In terms of spatial distribution, county carbon emissions in Gansu Province show a spatial pattern of "high in the middle and low at both ends". The carbon emissions in Longzhong region are significantly higher than other regions, and provincial capitals, industrial cities and prefecture-level cities are much higher than general county units. And with the change of time, the center of gravity of carbon emission distribution moved slightly to the southeast direction, but the overall change was little. Meanwhile, County carbon emissions have obvious spatial positive correlation. The distribution range of high-high clusters area is relatively stable, mainly concentrated in Lanzhou City and Baiyin City in Longzhong region, while the low-low clusters area is widely distributed, mainly distributed in southern ethnic region and Longnan City in Longdongnan region. In addition, LISA spatio-temporal transition matrix shows that there is a high path locking effect in county carbon emissions.
From the analysis of driving factors, we can see that economic scale, industrial structure and population size are the dominant factors for the spatial differentiation of county carbon emissions in Gansu Province, among which economic scale has the strongest explanatory power, and the explanatory power is significantly enhanced after interaction with other factors. The leading factors of the spatial differentiation of carbon emission levels in different regions are quite different. The market scale and carbon emission intensity have a significant impact on Hexi region and southern ethnic region, while the economic scale has a significant impact on Longzhong region and Longdongnan region.

Reviewer 2 Report
The article entitled “Research on Spatial and Temporal Pattern Evolution and Driving Factors of County Carbon Emissions in underdeveloped Regions: Gansu Province of Western China as an example” is written very well and according to the journal's scope. It is publishable in Climatic Change after addressing major revisions.
1. Please start the abstract with the main objectives of the study.
2. The results in abstract should write in a smooth paragraph without numbering.
3. Please write empirical findings in the abstract.
4. Write a study's main limitation at the end of the abstract.
5. It would be great if you include the study's main research questions.
6. Please write the main contributions of the study.
7. Please include the validation of models used in the study to approach study objectives.
8. Study’s limitations and future research should write at the end of the conclusion section.
9. The second statement of the introduction is without any justification. Therefore, it is strongly recommended to the authors update the statement with the given studies [1-3] as “In response to climate change, 60% of countries or regions in the world are implementing carbon neutral strategies in different ways [1-3]”
[1] Climate change policies of the four largest global emitters of greenhouse gases: their similarities, differences and way forward
[2] Extreme weather events risk to crop-production and the adaptation of innovative management strategies to mitigate the risk
[3] Estimating smart energy inputs packages using hybrid optimisation technique to mitigate environmental emissions of commercial fish farms
Author Response
Response: Thanks for the reviewers for their confirmation and comments. Based on your opinion, we have answered your questions one by one.
Firstly, based on your suggestions 1, 2, 3, 4, 5, 6 and 7, we further refined and improved the summary of this paper, including the main problems, objectives, method models, findings, contributions and limitations of the research. The text is revised as follows:
Abstract: Revealing the spatio-temporal evolution pattern of county carbon emissions and its driving factors is of great significance for promoting ecological civilization and green transformation development in Gansu Province. In this paper, Gini coefficient, standard deviation ellipse and spatial autocorrelation model were used to analyze the overall changes, regional differences, spatio-temporal evolution pattern and clustering characteristics of carbon emissions in 87 counties of Gansu Province from 1997 to 2017, and the driving factors of carbon emissions were detected by geographic detectors. The results show that the county carbon emissions in Gansu Province showed an “first urgent and then slow” upward trend, and the difference of carbon emissions level has a slight decreasing trend and there are significant regional differences. Compared with other regions, the difference of county carbon emissions level in Longzhong region has a smaller decline. Meanwhile, the county carbon emissions show a spatial differentiation characteristics “medium-high and low-outside”, among which the carbon emissions in areas with better economic foundations are much higher than those in other areas, and the spatial polarization effect is obvious. In addition, there is a significant spatial positive correlation between county carbon emissions. The counties with high-high clusters is relatively stable, mainly concentrated in the Longzhong region, while counties with low-low clusters are slightly reduced, mainly concentrated in southern ethnic region and Longdongnan region, and the county carbon emission clusters type has a spatial locking effect. This is mainly due to the large differences in economic scale, industrial structure and population size in Gansu Province, and the interaction between economic scale and other factors has a more significant impact on the spatial differentiation of carbon emissions. Moreover, the leading influencing factors of county carbon emission differences also have regional differences. Therefore, differentiated and targeted carbon emission reduction strategies need to be implemented urgently.
Secondly, according to the requirements of the journal's submission format, the content structure distribution and the structure of previous references, this paper believes that writing the limitations of research and future research in the discussion section will make the article more coherent and logical. We look forward to your further review.
Thirdly, according to the experts' suggestions, we carefully studied the literature, revised the second sentence of the introduction and further cited relevant literature for evidence. The text is revised as follows:
As the increase of anthropogenic CO2 emissions is one of the key factors of global warming, carbon neutralization has always been a concern [2, 3]. Of course, in order to minimize the threat of global climate change, many countries are trying to reduce the harmful effects of increased carbon emissions by signing the Paris Agreement and implementing energy conservation and emission reduction policies [4]. However, in recent years, climate warming caused by greenhouse gas emissions, such as carbon dioxide, has increasingly aggravated the deterioration of the global ecological environment, and has also accelerated the spread of diseases to a certain extent [5]. At the same time, the international community also recognizes that the climate problem is no longer a regional problem, nor can it be solved by a certain country or region, but a global problem that all countries in the world must deal with together, and it is necessary to adopt collective, coordinated and differentiated countermeasures to reduce the current greenhouse gas emissions [6, 7].
- Broadstock, D.; Ji, Q.; Managi, S.; Zhang, Y. Pathways to carbon neutrality: Challenges and opportunities.Resour Conserv Recy. 2021, 169, 105472.
- Iqbal, ; Abbasi, K.R.; Shinwari, R.; Wan, G.C.; Ahmad, M.; Tang, K. Does exports diversification and environmental innovation achieve carbon neutrality target of OECD economies? J. Environ. Manage. 2021, 291, 112648.
- Huang, M,T.; Zhai, P.M. Achieving Paris Agreement temperature goals requires carbon neutrality by middle century with far-reaching transitions in the whole society. Adv Clim Chang Res. 2021, 12(2): 281-286.
- Meehl, G.A.; Teng, H.; Arblaster, J.M. Climate model simulations of the observed early-2000s hiatus of global warming. Nat Clim Change. 2020, 4(10): 898-902.
- Codal, K.S.; Ari, I.; Codal, A. Multidimensional perspective for performance assessment on climate change actions of G20 countries. Environ Dev. 2021, 39, 100639.
- Chancel, L. Global carbon inequality over 1990– Nat Sustain. 2022, 5, 931-938.

Reviewer 3 Report
This paper selects 87 counties in Gansu Province as research objects, and analyzes the overall changes, regional differences, temporal and spatial changes and clustering characteristics of carbon emissions. The research idea is relatively clear and the perspective is relatively new, but there are still the following shortcomings:
(1) At present, there are many researches on carbon emissions, and the research methods and systems are very mature. The perspectives include regional, national, urban and even county-level scales. Therefore, what is the difference between the analysis on the evolution rules and influencing factors of carbon emissions from the perspective of county and previous studies? The innovation and necessity of the research are not specifically described in the article, so the innovation and necessity of the research can be added in the article.
(2) It is mentioned in 2.3.2 and 3.3 that five time point data in 1997, 2002, 2007, 2012 and 2017 are selected to measure the spatial movement characteristics of county carbon emissions in Gansu Province with the help of standard deviation ellipses. Has the article considered the problem of model suitability? Is it verified that the model can characterize the external gas space flow analysis of carbon emissions?
(3) The paper deeply analyzes the impact of 9 factors, such as population, economy, investment and technology, on carbon emissions. However, the specific rules and rationality of indicator selection are not stated. Please provide additional information?
(4) It is suggested to modify the format of figures and tables. As shown in Figure 2, the regional differences of carbon emissions in Gansu Province cannot be seen intuitively. At the same time, please check the full text carefully, there are some small mistakes in the English and Chinese expressions.
Author Response
Response: Thank the experts for their comments. Based on expert opinions, we answered your questions one by one.
Firstly, based on your suggestions, we further improved the innovation and necessity of this study in the introduction from the perspective of the county. The revised contents are as follows:
County, as the most basic administrative unit in China, accounts for 78% of China's land area, 71.94% of the population and 51.80% of the GDP, which is the basic spatial unit and carrier of economic development and industrial transfer, and the main battleground for future urbanization development [2]. Compared with provinces and cities, counties can better capture regional heterogeneity, which is of great significance for the adjustment and transformation of economic structure and development mode, and actively promotes the realization of carbon emission reduction targets [3]. Thus, strengthening research on carbon emissions at county level is conducive to improving the scientificity, pertinence and operability of energy conservation and emission reduction measures [4,5].
However, most carbon emission studies based on traditional statistical data are often limited to the national or provincial level due to data limitations, and it is difficult to refine to the county scale, which cannot provide more strong support for formulating regional and differentiated carbon emission reduction policies; Especially in Gansu Province, which is rich in resources and energy, the county is not only an important part of the industrial zone and a contributor to carbon emissions, but also a key administrative unit to implement the goal and policy of "double carbon". However, the research on carbon in the western region is relatively weak. Therefore, it is very necessary to study the county carbon emission in Gansu Province which is located in the western underdeveloped area. At the same time, the current use of night light data to estimate carbon emissions is mainly based on DMSP/OLS data, and the research time is concentrated before 2013, so it is difficult to dynamically monitor and track the development trend of carbon emissions in recent years. However, since 2012, NPP/VIIRS data is quite different from it in terms of spatial resolution, pixel brightness value and other data characteristics, which has become an obstacle and bottleneck for dynamic estimation and monitoring of regional carbon emissions in a long time series. The CEAD night light inversion carbon emission data used in this paper provides a more accurate data basis for district and county scale research, and its data has continuity and timeliness. In addition, because of differences in urbanization level and economic foundation, the influencing factors of carbon emissions exist variation. Although large-scale research can control the overall situation, it is not conducive to exploring different development stages and regionally targeted differentiated control. Common but differentiated responsibilities for carbon reduction should be reflected across regions. Gansu Province, as an important ecological hub area in the west and the whole country, is driven by rapid industrialization and urbanization, and its carbon emission reduction work is imminent. Thus, though spatial analysis function of ArcGIS, Gini coefficient, standard deviation ellipse, and spatial autocorrelation methods, this paper aims to analyze the overall and regional spatio-temporal characteristicsof carbon emissions of 87 counties in Gansu Provinc, and further use geographic detectors to explore the impact of social and economic factors on carbon emission, in order to promote the low-carbon, green and high-quality development of Gansu Province.
Secondly, according to the expert opinions, we have further understood the method principle of this model and its applicability to the analysis in this paper while supplementing the standard deviation ellipse formula in this paper. And make the following explanation:
The standard deviation ellipse (SDE) was proposed by D. Welty Lefever, a sociology professor at the University of Southern California in 1926, which is also called Lefever's "Standard Deviation Ellipse" (Lefever's directional distribution). It is a classical algorithm that can analyze the direction and distribution of points at the same time. Any event can be abstracted as a point in space. Then, as the basis of all forms, points also have their attribute characteristics. For example, whether the distribution of points is uniform, random or clustered, and the number, direction and range of points, these attribute characteristics are related to point data. Therefore, this method is used to reveal the spatial distribution characteristics of various geographical elements, and has been widely used in society, demography, criminology, geology, ecology and other fields. In other words, the biggest feature of this method is to measure the direction and distribution of a group of data. This is also one of the purposes of using it in this paper, that is, to reveal the spatial distribution characteristics of carbon emissions in different counties. Without considering other impacts of external flows, the carbon emissions of the whole Gansu Province are composed of carbon emissions of different county units, but the amount of carbon emissions of each county is different, which leads to spatial heterogeneity of the distribution characteristics of carbon emissions in Gansu Province. In order to reveal the spatial distribution characteristics of carbon emission elements in Gansu Province from multiple perspectives, such as identifying the spatial distribution center, direction and scope, it is found that the model has been applied more maturely and achieved fruitful results in the research of natural gas, carbon emissions, etc. on the basis of reference to previous studies. Therefore, this paper analyzes the evolution characteristics of carbon emissions in Gansu Province by quoting the model, and the research results are basically consistent with the actual situation. This also proves the applicability of the model to this paper, which can directly reflect the spatial aggregation characteristics of carbon emissions and their changes with time.
Thirdly, according to expert opinions, we have supplemented the reasons for the selection of driving factor indicators in 2.3.4. The article is revised as follows:
Referring to the existing research, following the principles of availability of county data and scientific and typical construction of indicator system, this paper focuses on the investigation and in-depth analysis of the impact of 9 factor including population, economy, investment and technology on carbon emissions (Table 1). Studies have confirmed that population is one of the main driving factors of carbon emissions [54-56], and population has an impact on carbon emissions through production and consumption behavior and has a two-way effect. Compared with population density, structure and other indicators, the population size and urbanization level selected in this paper can better reflect the direct impact of population changes on carbon emissions in the process of urbanization; The promotion of regional economy makes changes in production and consumption patterns, and then changes the level of carbon emissions. Therefore, this paper selects economic scale and financial input to characterize the level of affluence that affects environmental pressure; Because the irrational industrial structure is the contributing factor of China's carbon emissions, and the economy of Gansu Province is generally backward, most counties are still in the middle stage of industrialization development, so the ratio of the added value of secondary production to the added value of primary production is used to discuss the impact of industrial structure on county carbon emissions; Consumption is an important carriage for urban economic development, and the improvement of county disposable income will promote the increase of urban consumption emissions [57]. Therefore, this paper selects the per capita disposable income of urban residents to represent the impact of living standards on carbon emissions; The market potential of large-scale and high consumption attracts the migration of manufacturing and service industries, which will have an impact on carbon emissions. Therefore, this paper selects market scale and investment level to characterize the impact on carbon emissions; Industrialization plays a decisive role in reducing carbon emissions, and the improvement of energy efficiency is one of the key means to achieve carbon emission reduction [58]. This paper selects carbon emission intensity to characterize the impact of technological progress on carbon emissions.
- Shi, A. The impact of population pressure on global carbon dioxide emissions, 1975-1996: Evidence from pooledcross-country data. Ecol. Econ. 2003, 44(1): 29-42.
- O'Neill, B.C.; Liddle, B.; Jiang, L.W.; Smith, K.R.; Pachauri, S.; Dalton, M.; Fuchs, R. Demographic change and carbon dioxide emissions. Lancet. 2012, 380(9837): 157-164.
- Zheng, X.Q.; Lu, Y.L.; Yuan, J.J.; Baninla, Y.; Zhang, S.; Stenseth, N.C.; Hessen, D.O.; Tian, H.Q.; Obersteiner, M.; Chen, D.L. Drivers of change in China's energy-related CO2 PNAS. 2020, 117(1): 29-36.
- Duan, C.C.; Zhu, W.J.; Wang, S.G.; Chen, B. Drivers of global carbon emissions 1990–2014. J. Clean. Prod. 2022, 371,
- Ren, H.M.; Gu, G.F.; Zhou, H.H. Assessing the low-carbon city pilot policy on carbon emission from consumption and production in China: how underlying mechanism and spatial spillover effect? Environ. Sci. Pollut. Res. 2022, 29(47): 71958-71977.
Fourth, according to expert opinions, we further adjusted and revised the format of the chart in this paper, carefully checked the content of the full text, and improved some sentences that were not properly expressed.

Reviewer 4 Report
The paper is interesting, but it needs to be improved further.
Main remarks
Abstract-It needs to be more synthetic. It's too descriptive in the results.
Introduction
Despite having relevant references please consider adding more ot international level.
2.3.2. Standard Deviation Ellipse-You should add the formulae of Yang et al. and not only mentioning it.
4.2. Driving Factors of Carbon Emissions in Different Regions-This subsection needs to be more detailed.
Author Response
Response: Thanks for the reviewers for their confirmation and comments. Based on your opinion, we have answered your questions one by one. The revised part of the text has been marked in blue font.
Firstly, We have improved the abstract to reflect its comprehensiveness and depth. The revised contents are as follows:
Abstract: Revealing the spatio-temporal evolution pattern of county carbon emissions and its driving factors is of great significance for promoting ecological civilization and green transformation development in Gansu Province. In this paper, Gini coefficient, standard deviation ellipse and spatial autocorrelation model were used to analyze the overall changes, regional differences, spatio-temporal evolution pattern and clustering characteristics of carbon emissions in 87 counties of Gansu Province from 1997 to 2017, and the driving factors of carbon emissions were detected by geographic detectors. The results show that the county carbon emissions in Gansu Province showed an “first urgent and then slow” upward trend, and the difference of carbon emissions level has a slight decreasing trend and there are significant regional differences. Compared with other regions, the difference of county carbon emissions level in Longzhong region has a smaller decline. Meanwhile, the county carbon emissions show a spatial differentiation characteristics “medium-high and low-outside”, among which the carbon emissions in areas with better economic foundations are much higher than those in other areas, and the spatial polarization effect is obvious. In addition, there is a significant spatial positive correlation between county carbon emissions. The counties with high-high clusters is relatively stable, mainly concentrated in the Longzhong region, while counties with low-low clusters are slightly reduced, mainly concentrated in southern ethnic region and Longdongnan region, and the county carbon emission clusters type has a spatial locking effect. This is mainly due to the large differences in economic scale, industrial structure and population size in Gansu Province, and the interaction between economic scale and other factors has a more significant impact on the spatial differentiation of carbon emissions. Moreover, the leading influencing factors of county carbon emission differences also have regional differences. Therefore, differentiated and targeted carbon emission reduction strategies need to be implemented urgently.
Secondly, for the introduction part, we have further supplemented and improved the relevant research based on expert opinions, and correspondingly added more internationally representative literature. The specific modifications are as follows:
As the increase of anthropogenic CO2 emissions is one of the key factors of global warming, carbon neutralization has always been a concern [2, 3]. Of course, in order to minimize the threat of global climate change, many countries are trying to reduce the harmful effects of increased carbon emissions by signing the Paris Agreement and implementing energy conservation and emission reduction policies [4]. However, in recent years, climate warming caused by greenhouse gas emissions, such as carbon dioxide, has increasingly aggravated the deterioration of the global ecological environment, and has also accelerated the spread of diseases to a certain extent [5]. At the same time, the international community also recognizes that the climate problem is no longer a regional problem, nor can it be solved by a certain country or region, but a global problem that all countries in the world must deal with together, and it is necessary to adopt collective, coordinated and differentiated countermeasures to reduce the current greenhouse gas emissions [6, 7].
- Broadstock,D.;Ji, Q.; Managi, S.; Zhang, D.Y. Pathways to carbon neutrality: Challenges and opportunities.Resour Conserv Recy. 2021, 169, 105472.
- Iqbal, N.; Abbasi, K.R.; Shinwari,R.;Wan, G.C.; Ahmad, M.; Tang, K. Does exports diversification and environmental innovation achieve carbon neutrality target of OECD economies? J. Environ. Manage. 2021, 291, 112648.
- Huang, M,T.; Zhai, P.M.Achieving Paris Agreement temperature goals requires carbon neutrality by middle century with far-reaching transitions in the whole society.Adv Clim Chang Res. 2021, 12(2): 281-286.
- Meehl, G.A.; Teng, H.; Arblaster, J.M. Climate model simulations of the observed early-2000s hiatus of global warming. Nat Clim Change. 2020, 4(10): 898-902.
- Codal, K.S.; Ari, I.; Codal, A. Multidimensional perspective for performance assessment on climate change actions of G20 countries. Environ Dev. 2021, 39, 100639.
- Chancel, L. Global carbon inequality over 1990–2019. Nat Sustain.2022, 5, 931-938.
Thirdly, we have supplemented the formula of standard deviation ellipse in 2.3.2. The specific supplementary contents are as follows:
, (2)
,. (3)
(4)
In the formula, and are the central coordinates of the ellipse respectively; xi and yi are the coordinates of the i-th study unit, respectively; SDEx and SDEy are the variances of the x axes and y axes of the ellipse, respectively; tanθ is the rotation angle of the ellipse; and are the deviation between the center of the ellipse and the center of the i-th element space respectively.
Fourthly, based on expert opinions, we made a detailed analysis of 4.2 driving factors of carbon emissions in different regions. The revised content is as follows:
Table 3 and Figures 6(b,c,d,e) show that the influencing factors of spatial differentiation of county carbon emission levels in different regions are heterogeneous.
The leading factors of carbon emission in Hexi region are market size (0.855), economic size (0.804) and economic size ∩ investment level (0.999). Hexi Corridor, as an important military town in the northwest of China and an indispensable part of the ancient Silk Road, is rich in resources and energy. In its early days, it was a rich place for military strategists to compete. The urban function of "tea and horse trade" has stimulated the expansion of trade and consumption markets in this region, and the constant increase of social fixed investment. In addition, the rapid development of resource-based cities such as Yumen, Jinchuan, Jiayuguan and Suzhou has injected vitality into the economy. However, the rapid development and extensive utilization of resources and energy will inevitably promote the increase of carbon emissions. With the depletion of resources in recent years, it’s industry is facing the challenge of adjustment and transformation, and it is urgent to promote the development of local economy to green environment.
The leading factors of carbon emission in Longzhong region are economic size (0.768), investment level (0.707) and population size ∩ investment level (0.999). Lanzhou and Baiyin, as important industrial towns in the west, economic centers in the upper reaches of the Yellow River and important cities in the Silk Road Economic Belt, have a fast flow of regional factor resources and strong regional radiation. The economic foundation of the counties in this region is better than that of other regions as a whole. With the further promotion of the construction of Lanzhou-Xining urban agglomeration, the market radiation scope is gradually expanding, production factors are concentrated, population concentration is high, and the scale of construction land is expanding. However, due to the terrain limitation of the valley basin, the dense population and production activities have increased the atmospheric environmental pressure, making it a high-value concentration area of carbon emissions. In addition, heavy industries with high water consumption, high energy consumption and high pollution, such as petrochemical industry and nonferrous metal smelting, still account for a large proportion in the region. The convergence of economic structure is common, and it is difficult to change the transformation and upgrading of economic industrial structure and the conversion of new and old kinetic energy in the short term. While the high energy consuming industries promote economic development, they also contribute to the increase of carbon emissions in the region.
The dominant factors of carbon emission in Longdongnan region are economic size (0.797), living standard (0.686) and economic size ∩ carbon emission intensity (0.997). The region has laid a solid foundation for the rapid promotion of county economy by virtue of its favorable location advantages and good industrial foundation such as electronics, electricians and equipment manufacturing. However, traditional industries still account for a certain proportion, and green manufacturing with technological innovation as the core is relatively limited. With the improvement of the living standard of urban residents, the concept of urban consumption has gradually changed, and the consumption of various goods and services has increased. The market potential of large-scale and high consumption has attracted the migration of manufacturing and service industries, and the living standard has increased the consumption carbon emissions. In addition, the region has a large national energy and chemical industry base. Pingliang and Qingyang have achieved rapid economic growth by virtue of their energy endowment advantages, and have long been at a higher level in Gansu Province. However, the energy structure dominated by coal has promoted the agglomeration of high energy consuming industries, which is not conducive to the realization of the "low carbon" and "green" development goals in the region.
The main influencing factors of carbon emission in southern minority region are carbon emission intensity (0.844), population size (0.816) and economic size ∩ carbon emission intensity (1.00). This region is an arid and alpine agricultural and pastoral area in the intersection of Gansu and Qinghai. The ecosystem is fragile, the industrial foundation is weak, and investment is limited. Agriculture and animal husbandry are the main channels of the county economy. Higher livestock breeding scale, while increasing income, also leads to increasing carbon emissions. At the same time, restricted by natural environment, economic foundation, traditional ways of production and living and other energy inputs, this region has extensive household energy consumption structure and low energy utilization efficiency. The increase of population and the development of urbanization promote the increasing energy consumption of buildings and living, which leads to the increase of carbon emissions from production and living.

Round 2
Reviewer 2 Report
The authors have completely ignored the comments and did not address them properly. All the comments have not been addressed.
1. Please start the abstract with the main objectives of the study.
Abstract did not start with main objectives of the study.
3. Please write empirical findings in the abstract.
No empirical findings have written in the article.
4. Write a study's main limitation at the end of the abstract.
Authors did not address this comment.
5. It would be great if you include the study's main research questions.
Research questions have not written in the introduction.
6. Please write the main contributions of the study.
Main contributions of articles have not written.
7. Please include the validation of models used in the study to approach study objectives.
Authors did not provide any validation of the models.
8. Study’s limitations and future research should write at the end of the conclusion section.
The section of Study’s limitations and future research has not written after conclusion.
9. The second statement of the introduction is without any justification. Therefore, it is strongly recommended to the authors update the statement with the given studies [1-3] as “In response to climate change, 60% of countries or regions in the world are implementing carbon neutral strategies in different ways [1-3]”
[1] Climate change policies of the four largest global emitters of greenhouse gases: their similarities, differences and way forward
[2] Extreme weather events risk to crop-production and the adaptation of innovative management strategies to mitigate the risk
[3] Estimating smart energy inputs packages using hybrid optimisation technique to mitigate environmental emissions of commercial fish farms
Although I recommend modifying the above statement, the authors have tried their best to remove it from the introduction.
I am not in favor of publishing this article until all comments will not be addressed.
Author Response
Thank you very much for your comments and suggestions. Maybe we didn't fully understand the meaning of your comments during the first revision, so there are some defects in the revision process. Please forgive me. Based on this, we have carefully revised and improved your comments again.
- Please start the abstract with the main objectives of the study.
- Please write empirical findings in the abstract.
- Write a study's main limitation at the end of the abstract.
Response: Firstly, comments 1, 3 and 4 are all about abstract problems. We have revised the abstract according to your suggestions. The first sentence is the research objective, followed by the empirical results of the study. The main limitations of the study are also supplemented at the end of the summary. The modifications are as follows:
Abstract: This paper used Gini coefficient, standard deviation ellipse and spatial autocorrelation model to analyze the overall changes, regional differences, spatio-temporal evolution pattern and clustering characteristics of carbon emissions in 87 counties in Gansu Province from 1997 to 2017, based on which driving factors of carbon emissions were detected using the geographic detector model, so as to provide a reference for promoting low-carbon green development and ecological civilization construction in Gansu Province. The empirical research results found that county carbon emissions in Gansu Province showed an upward trend “first urgent and then slow” from 1997 to 2017, and the difference of carbon emissions level has a slight decreasing trend and there are significant regional differences. Compared with other regions, the difference of county carbon emissions level in Longzhong region has a smaller decline. Meanwhile, county carbon emissions exsit spatial differentiation characteristics “medium-high and low-outside”, among which the carbon emissions in areas with better economic foundations are much higher than those in other areas, and the spatial polarization effect is obvious. In addition, there is a significant spatial positive correlation between county carbon emissions. The counties with high-high clusters is relatively stable, mainly concentrated in the Longzhong region, while counties with low-low clusters are slightly reduced, mainly concentrated in southern ethnic region, and the county carbon emission clusters type has a spatial locking effect. This is mainly due to the large differences in economic scale, industrial structure and population size in Gansu Province, and the interaction between economic scale and other factors has a more significant impact on the spatial differentiation of carbon emissions. Moreover, the leading influencing factors of county carbon emission differences also have regional differences. Therefore, differentiated and targeted carbon emission reduction strategies need to be implemented urgently. Due to the lack of real county energy consumption statistics, the research results need to be further tested for robustness.
- It would be great if you include the study's main research questions.
Response: Thanks for the experts' opinions. We explained the main problems of the study in lines 154-162 of the introduction. The modifications are as follows:
Based on this, this paper takes the carbon emissions of 87 counties in Gansu Province as the research object, and with the support of ArcGIS spatial analysis function, Gini coefficient, standard deviation ellipse, spatial autocorrelation and other methods, explores what is the temporal and spatial change trend of carbon emissions of counties in Gansu Province? What are the characteristics or laws of evolution? What factors affect its carbon emission level? The purpose is to reveal the difference of the implementation effect of energy conservation and emission reduction measures under different carbon emission levels, and provide a theoretical basis for the implementation of phased carbon emission reduction measures.
- Please write the main contributions of the study.
Response: Thank you for your comments. We have supplemented the main contributions of the article in the conclusion. The supplementary contents are as follows:
In conclusion, this paper systematically studied the temporal and spatial characteristics of carbon emissions at county level in Gansu Province, mainly reflected in: on the sample data, the carbon emissions data retrieved from CEADs nighttime lighting used in this paper provide a more accurate data basis for regional and county-level scale research, and fill the gap in energy data of statistical data at the micro scale; In terms of research content, this paper can reveal the internal structural characteristics of the temporal and spatial differences of carbon emissions in a more detailed way from the multi-dimensional perspective of time and space, as well as from the overall, regional and county multi-scale research of Gansu Province; In terms of research value, a comprehensive investigation of the temporal and spatial characteristics of carbon emissions in Gansu Province will help to clarify the sources of spatial differences in carbon emissions in Gansu Province, clarify the geographical scale of carbon emission reduction potential in Gansu Province, and provide a reference for formulating carbon emission reduction action plans.
- Please include the validation of models used in the study to approach study objectives.
Response: Thanks for the expert advice. The models we use in our research are analyzed after validation. Spatial autocorrelation model, for example, refers to the potential interdependence of observed data in the same distribution area of some variables. When using this model for analysis, we first tested the carbon emission data, and found that the standardized statistic Z values were all significant at the 1% level (P<0.01), and Moran's I were all greater than 0, indicating that the carbon emission of county units in Gansu Province showed a significant spatial positive correlation, and there was a high value (low value) agglomeration. This is also supplemented in the article. Before using the geographic detector model to analyze the driving factors of carbon emissions, we first used SPSS software to conduct KMO and Bartleet tests on the driving factors, and calculated that the KMO value was greater than 0.5, and the Bartleet spherical test results were all significant at the 1% level, indicating that the variables could be factor analyzed. In addition, the detection results of the driving factors of carbon emissions using the geographical detection model are roughly consistent with the actual situation in the study area, which also proves from the side that the model used in the study can better reflect the accuracy of the research results. These contents are also supplemented in the article. The details are in lines 280-283 and 382-387 of the article.
- Study’s limitations and future research should write at the end of the conclusion section.
Response: Thanks for the expert opinions, we supplemented the limitations of this study and the future research at the end of the conclusion. The supplementary contents are as follows:
However, due to the lack of real county level energy consumption statistics and the latest county level carbon emission data, some studies cannot be carried out or the results of the studies vary greatly or are not accurate enough. Therefore, it is recommended that the county level government of Gansu Province establish an energy consumption statistics system, develop an energy balance sheet, and directly obtain carbon emission data to provide a basis for formulating more effective county-level carbon emission reduction policies. In addition, in future research, in addition to nighttime lighting data, multidimensional data such as land, socio-economic and physical geography can be taken into account when calculating carbon emissions, which can not only expand the application scope of nighttime lighting data sets, provide reference for governments and policy makers, but also help to achieve coordinated emission reduction between different sectors.
- The second statement of the introduction is without any justification. Therefore, it is strongly recommended to the authors update the statement with the given studies [1-3] as “In response to climate change, 60% of countries or regions in the world are implementing carbon neutral strategies in different ways [1-3]”.
Response: Thank you for your comments and suggestions. Due to our misunderstanding of some of your comments, our modification is not perfect. In view of this, we have made an original supplement to the second sentence of the introduction according to your suggestion, and cited the specific research you suggested for support. Modify as follows:
In response to climate change, 60% of countries or regions in the world are implementing carbon neutral strategies in different ways [2-4]. Most countries, represented by the United States, the United Kingdom, the European Union and Japan, have set the carbon neutral time at 2050, which means the world consensus on carbon emission reduction is gradually forming [5-7].
- Smith, E.K.; Mayer, A. A social trap for the climate? Collective action, trust and climate change risk perception in 35 countries. Global Environ Chang. 2018, 49, 140-153.
- Ehsan, E.; Zainab, K. Muhammad Zubair Tauni, Hongxia Zhang, Xing Lirong. Extreme weather events risk to crop-production and the adaptation of innovative management strategies to mitigate the risk: A retrospective survey of rural Punjab, Pakistan. Technovation. 2022, 117, 102255.
- Ehsan, E.; Zainab, K. Estimating smart energy inputs packages using hybrid optimisation technique to mitigate environmental emissions of commercial fish farms. Appl. 2022, 326, 119602.
- Broadstock, D.; Ji, Q.; Managi, S.; Zhang, Y. Pathways to carbon neutrality: Challenges and opportunities. Resour Conserv Recy. 2021, 169, 105472.
- Iqbal, ; Abbasi, K.R.; Shinwari, R.; Wan, G.C.; Ahmad, M.; Tang, K. Does exports diversification and environmental innovation achieve carbon neutrality target of OECD economies? J. Environ. Manage. 2021, 291, 112648.
- Huang, M,T.; Zhai, P.M. Achieving Paris Agreement temperature goals requires carbon neutrality by middle century with far-reaching transitions in the whole society. Adv Clim Chang Res. 2021, 12(2): 281-286.

Reviewer 3 Report
In addition to the impact of spatial scale scaling on data, it also has an impact on elemental relationships. In this paper, need to explain the essential differences between county scale studies and urban scale, grid scale, etc. on carbon emission related relationships, otherwise there is still a problem of insufficient scientific contribution.
In addition, although counties are smaller administrative units than cities, they do not have administrative decision-making power and generally do not have independent industrial access and exit policies, so why study the driving mechanisms of carbon emissions at the county scale? Further refinement is still needed.
Author Response
Thank the reviewers for their careful guidance and valuable comments on my paper. Through careful study of the literature and discussion of the team, this article further explains your questions.
In addition to the impact of spatial scale scaling on data, it also has an impact on elemental relationships. In this paper, need to explain the essential differences between county scale studies and urban scale, grid scale, etc. on carbon emission related relationships, otherwise there is still a problem of insufficient scientific contribution.
Response: The research on carbon emissions at the provincial and municipal levels is to consider the development of the whole province or the whole city from the overall perspective. Although it plays a leading role in overall planning to a certain extent, the basic carrier is still the county. Because the county economy has changed from small to large, from weak to strong, and from strong to excellent, it also lays a foundation for the development and growth of cities and provinces. On the other hand, without the overall planning and leading role of provinces and cities, county development is also difficult to achieve. Therefore, the study of county scale and city scale is indispensable. They complement, promote and influence each other. Only by realizing the combination and integration of the two, forming an integrated whole, and making sure that we have each other, can we make contributions to the carbon emission reduction goals of the whole province and even the whole country.
Through literature research, we found that most of the current carbon emission studies focused on the provincial, municipal and grid levels, basically using the top-down method to obtain data. Although the hot areas of carbon emissions have been explained, there is no strong explanation for the characteristics of smaller county carbon emissions. The research also further shows that the spatial characteristics of carbon emissions at the provincial, municipal, grid and county scales are significantly different and have significant spatial correlation, which means that the realization of the "double carbon" goal cannot be achieved by "fighting alone". Adjacent regions should carry out joint actions against carbon emissions, but carbon emissions research at the provincial, municipal and grid scales has limited guiding significance for low-carbon actions at the district and county levels, Therefore, it is necessary to supplement and refine the regional carbon emission data with smaller spatial dimensions, and the county mainly undertakes the function of "re implementation" of indicator placement and spatial implementation. Compared with provincial and municipal scales, the county carbon emission research is more conducive to the realization of strategic objectives. At the same time, the different problems faced by each county lead to different carbon emission reduction problems. The county level carbon emission research focuses on starting from the existing practical problems in the county and anchoring the direction and severity of the county level emission reduction problems, so as to clarify the focus of carbon control and emission reduction at the municipal and provincial levels.
In addition, although counties are smaller administrative units than cities, they do not have administrative decision-making power and generally do not have independent industrial access and exit policies, so why study the driving mechanisms of carbon emissions at the county scale? Further refinement is still needed.
Response: At the administrative level, although the county is smaller than the city, the county includes municipal districts, county-level cities, counties, autonomous counties and other units, and the county also has some administrative decision-making power and some independent industrial access and exit policies. In recent years, with the development of market economy, the drawbacks of the system of city governing county in some regions are increasingly obvious, such as low administrative efficiency and limited urban radiation. In order to stimulate the development vitality of the basic unit in the county area and build a new pattern of urbanization development, the General Office of the CPC Central Committee and the General Office of the State Council also issued the Opinions on Promoting the Urbanization Construction with the County as an Important Carrier, proposing to promote the urban-rural integration development with the county as the basic unit, giving play to the role of the county in connecting the city and serving the countryside, and giving classified guidance to the development of the county. As an important part of the urban system and a key support for the integrated development of urban and rural areas, county towns play an important role in urban and provincial construction.
At present, there are many studies on carbon emissions and influencing factors at different scales. Most studies believe that resource endowment, per capita energy output, the proportion of high energy consuming industries in industrial output value and the proportion of coal in fossil energy consumption are the main influencing factors affecting the differences in carbon emission intensity among regions, provinces and cities. However, the social and economic development level, resource and environmental characteristics, carbon emission characteristics, governance potential level and development orientation of each region are very different, and emission reduction policies should also be different. The analysis and policy recommendations from the provincial or municipal level may not be consistent with the actual situation of most counties, and it is not conducive to the specific implementation of the counties. The "one size fits all" municipal low carbon governance scheme and policy constraints are obviously not good, taking into account the country The overall emission reduction strategy at provincial and municipal levels and the decomposition idea of differentiated targets and indicators for their own development are important points to achieve carbon emission reduction. And the research also proves that the county carbon emissions are more closely related to social and economic development and other factors. In addition, the impact mechanism of carbon emissions also has a spatial effect, and the spatial difference of carbon emissions is the basis for international discussions on the decomposition of emission reduction targets. The spatial difference of domestic carbon emissions is not only the basis for the decomposition of emission reduction tasks, but also the basis for financial transfer payments. As a complete grass-roots unit in China's administrative system, the county scale is the best scale for the sustainable use, management and planning of resources in China. Understanding the research on carbon emission differentiation mechanism at the county level is of great practical significance for grassroots governments to implement emission reduction tasks more specifically and operationally.
At the same time, as the national ecological environment base and basic economic unit, the county's carbon emissions account for more than 50% of the national total, and its economic total accounts for more than 60% of the national total. Driven by urbanization and industrialization, the county began to suffer from energy structure imbalance, low resource utilization rate, insufficient ecological background protection and other problems in exchange for rapid economic growth with the reduction of environmental resource carrying capacity. At the same time, as a key level and basic governance unit for balancing urban and rural carbon and oxygen balance in China, the development characteristics of counties are different from those of large and medium-sized cities. First of all, the county economic growth mostly depends on the cost of cheap labor and natural resources. The extensive development model is one of the important reasons for the rapid growth of China's carbon emissions; Secondly, a large number of industrial and energy activities are concentrated in the county, and economic development has a stronger correlation impact mechanism on the change trend of carbon emissions. Analyzing the driving mechanism of carbon emissions from the perspective of the county is of practical significance for guiding the differential carbon control and emission reduction, and achieving carbon peak and carbon neutralization as soon as possible. Although this paper has studied the driving mechanism of carbon emissions at the county level, there is still a lack of comparative research on counties with different attributes. In order to promote the early realization of green and high-quality development of people's lives with carbon emission reduction goals, it is necessary to further strengthen the research on counties with different attributes or counties with significant carbon emissions in the future, so as to generate more valuable opinions and references.

Round 3
Reviewer 2 Report
All earlier comments have been resolved by the authors, and the manuscript is now suitable for publishing in its current form.
Reviewer 3 Report
Thanks to the authors for the in-depth discussion, the county study has some guiding significance and is recommended for acceptance.